# The Structure–Properties–Cytotoxicity Interplay: A Crucial Pathway to Determining Graphene Oxide Biocompatibility

**DOI:** 10.3390/ijms22105401

**Published:** 2021-05-20

**Authors:** Marta Dziewięcka, Mirosława Pawlyta, Łukasz Majchrzycki, Katarzyna Balin, Sylwia Barteczko, Martyna Czerkawska, Maria Augustyniak

**Affiliations:** 1Institute of Biology, Biotechnology and Environmental Protection, Faculty of Natural Sciences, University of Silesia in Katowice, Bankowa 9, 40-007 Katowice, Poland; sylwia.barteczko95@gmail.com (S.B.); martynaczerkawska88@gmail.com (M.C.); maria.augustyniak@us.edu.pl (M.A.); 2Department of Engineering Materials and Biomaterials, Faculty of Mechanical Engineering, Silesian University of Technology, Konarskiego 18A, 44-100 Gliwice, Poland; miroslawa.pawlyta@polsl.pl; 3Center for Advanced Technology, Adam Mickiewicz University, Uniwersytetu Poznańskiego 10, 61-614 Poznań, Poland; lukasz.majchrzycki@gmail.com; 4Institute of Physics, University of Silesia, 75 Pulku Piechoty 1A, 41-500 Chorzow, Poland; katarzyna.balin@us.edu.pl

**Keywords:** graphene oxide (GO), toxicity, biocompatibility, in vivo model, physicochemical properties

## Abstract

Interest in graphene oxide nature and potential applications (especially nanocarriers) has resulted in numerous studies, but the results do not lead to clear conclusions. In this paper, graphene oxide is obtained by multiple synthesis methods and generally characterized. The mechanism of GO interaction with the organism is hard to summarize due to its high chemical activity and variability during the synthesis process and in biological buffers’ environments. When assessing the biocompatibility of GO, it is necessary to take into account many factors derived from nanoparticles (structure, morphology, chemical composition) and the organism (species, defense mechanisms, adaptation). This research aims to determine and compare the in vivo toxicity potential of GO samples from various manufacturers. Each GO sample is analyzed in two concentrations and applied with food. The physiological reactions of an easy model *Acheta domesticus* (cell viability, apoptosis, oxidative defense, DNA damage) during ten-day lasting exposure were observed. This study emphasizes the variability of the GO nature and complements the biocompatibility aspect, especially in the context of various GO-based experimental models. Changes in the cell biomarkers are discussed in light of detailed physicochemical analysis.

## 1. Introduction

Carbon plays a crucial role in the environment. It is a raw material, energy source, and a component of all organisms. Therefore, we can assume that carbon materials are more environmentally friendly than inorganic materials [1]. Carbon allotropes are naturally occurring materials, and graphite has had no toxicity problems for hundreds of years. Graphene, a single graphite layer, is the common building element of other carbon structures and is valuable for technological purposes, mainly due to its excellent mechanical, electronic, and optical properties [2,3]. More recently, understanding graphene’s various chemical properties has facilitated its use in high-performance energy generation and storage devices [4]. The two-dimensional shape of graphene was expected to be harmless at mild concentrations and suitable for biomedical research. While pure graphene has limited processability, its derivatives are more universal, and they inspire interdisciplinary research fields due to their broad applicability [5]. The discovery of graphene’s properties accelerated research on graphene oxide (GO), becoming an independent research area.

Graphene oxide is considered a promising material for applications in many industries due to its excellent water processability, amphiphilicity, covalent and non-covalent surface functionalization, and ability to quench fluorescence [6,7,8,9,10,11]. GO is highly chemically active due to the partial coverage of its elemental planes and edges with various functional groups, mainly hydroxyl and epoxy, ketone, ester, organosulfur, and lactol structures in the dry state. When dispersed in polar solvents such as water, it can hydrolyze to form carboxylic acid groups or sulfate groups heavily modified with various chemicals, as described in the literature [12,13,14].

Since GO is practically not found in nature, it is obtained by modified oxidation processes using any graphite source, such as structured layer graphite, expanded graphite, graphite rod electrodes, kish graphite, and graphite foils [13]. The materials used for the synthesis are diverse in terms of flake size, defect density, and layering, making it difficult to fully control the obtained GO properties. The product is also influenced by the quantity and quality of the reagents used, the reaction temperature, and the post-reaction mixture purification precision [15,16,17]. Reagent contamination remains part of the GO, and post-synthesis treatment can change the surface chemistry, for example, break down the carbon network structure by breaking the C-C bond or modifying the flakes’ side dimensions during the sonication process by cracking the flakes at the grain boundary [13,18]. Excessive oxidation and high temperature can lead to the breakdown of the flakes and the formation of a more complex structure due to the release of CO_2_ [19,20]. The lattice defects and the functional groups’ quality at the edges of the flakes and aggregation potential may determine the GO’s properties and influence the toxicity directly or indirectly [21]. Under laboratory conditions, certain parameters, such as the degree of oxidation, temperature, and pH value, can be regulated by appropriate synthesis protocols [22,23]. However, post-functionalization of graphene in situ in the body is very difficult to predict and clarify.

For several years, scientists have been trying to assess the toxicity of GO in various experimental models, but the results do not always lead to clear and consistent conclusions [24]. Graphene oxide used in the research is synthesized by multiple methods and generally characterized. In GO-based nanocomposites study, the carrier’s physicochemical properties are not being discussed thoroughly. Determining nanocarriers’ toxicity standards requires an interdisciplinary approach and multidirectional research based primarily on in vivo models. Experimental models must consider concentration/dose and exposure time and consider the GO material’s physicochemical characteristics and its potential interactions in biological buffers [25]. Establishing the structure–properties–cytotoxicity relationship seems fundamental in developing a biocompatible therapeutic system and drawing clear conclusions about toxicity.

With the above in mind, in this study, we attempted to assess the short-term in vivo toxicity of four GO products from various manufacturers in the context of their detailed physicochemical analysis.

The samples were prepared in the same conditions and intentionally “coded” to focus on the physical–chemical–biological relationships rather than the material’s origin. Based on our several years of experience with GO, we used the *Acheta domesticus* model organism with known physiology and reactions under stress conditions [26,27,28,29,30,31,32,33,34]. The following hypotheses were verified to resolve the relationship between the chemical composition and morphology of GO samples and selected cytotoxicity markers at the cellular level:
Graphene oxide from various sources does not cause significant differences in cell viability, oxidative stress level, apoptosis level, and DNA damage in *A. domesticus* during the 10-day exposure. The effect of all types of GO is similar, and the materials do not differ significantly in terms of physicochemical properties and toxicity.The selected physicochemical properties of GO are significant, and the materials significantly differentiate cell viability, the level of oxidative stress, the degree of apoptosis, and DNA damage at various time points during 10 days of exposure in *A. domesticus.* This may be due to the GO flakes’ size, aggregation potential, suspension stability, degree of oxidation, or the number of surface defects.


## 2. Results

### 2.1. Properties of Graphene Oxide Nanoparticles

#### 2.1.1. Morphological and Structural Analysis of GO Samples

Analyzed samples were chosen to identify the main differences in the GO structure. Sample 2 (S2) and sample 3 (S3) represent well-defined flake-like GO structures (Figure 1(A2,A3)). In contrast, the S1 and S4 samples are highly aggregated materials. Sample S4 forms thick, block-like structures, while S1 tends three-dimensional aggregates of nanometric flakes (Figure 1(A1,A4)).

The contrast in the TEM images indicates different thicknesses of the observed structures. S2 and S3 flakes are mostly transparent, which means a lower thickness than with S1, especially with the thickest S4 sample (Figure 2(A1,B1,C1,D1)). A more detailed analysis by AFM shows (see Figure 1(B2,B3)) that both S2 and S3 are mostly single-layer GO flakes, with a typical height of 1.0 nm, which is in line with the typical height of a GO layer in dry conditions [35]. Comprehensive analysis for flakes size performed based on more than 400 flakes for each sample shows that the average flake area of S3 is about 2 µm^2^, while for S2, the average flake is 0.2 µm^2^. Contrary to the above, the AFM results of the S4 sample show the height of aggregates even in the range of 1 µm and tens of nanometers in the case of the S1 sample (Figure 1(B1,B4)). No additional contrast in HAADF images (Figure 2(A2,B2,C2,D2)) confirms that the tested material contains only light elements (carbon, oxygen). The bright bands are visible in Figure 2(B2,C2,D2) are related to the carbon layers’ orientation (diffraction contrast).

The most significant differences are visible in the electron diffraction patterns obtained using the selected area electron diffraction (SAED). S1 and S2 samples have the form of circles, the diameters of which correspond to distances of approximately 2.08 Å (the distance between the d101 planes in graphite) and 1.23 Å (the distance between the d110 planes in graphite) (Figure 2 (A3,B3)). This result suggests that the tested carbon materials’ planes do not form a three-dimensional crystal lattice but are arranged randomly to each other. The diffraction reflections of the S3 sample are more evident and do not form clear circles (Figure 2(C3,C4)). According to the RS and AFM results, this may indicate a stronger bond between the carbon layers or a limited number of carbon layers. For sample S4, the electron diffraction characteristic for crystalline graphite was obtained (Figure 2(D3)). On this basis, it can be concluded that this carbon material consists of nanometric three-dimensional graphitized crystallites (based on Raman spectroscopy, it is approximately 16 nm), which is also confirmed by the exemplary HRTEM image (Figure 2(D4)), which shows a dozen or so flat and parallel arranged carbon layers. In the S1 and S2 samples, the number of approximately parallel carbon layers can be comparable (Figure 2(A4,B4)). However, they are not flat, defective, and accidentally rotated against each other.

The first-order Raman spectra of the S1–S3 samples are similar. The shape of the spectrum of the S4 sample is different (Figure 3(A1)). The first-order spectrum of sample S4 resembles polycrystalline graphite. Moreover, it is the only sample with visible reflections in the second-order spectrum (Figure 3(A2,A3)). The first-order spectra for samples S1–S3 are similar to black carbon spectra. It means that S1–S3 presented a less ordered structure (than S4) and contained smaller and more damaged layers of carbon. The quantitative analysis of the Raman spectra of the tested materials showed approximately zero shares of the D4 band (the share of the D4 band for the tested materials can be neglected). The positions of the respective bands for tested samples are similar. The most significant differences concern the D3 band position, but this is probably due to this band’s large width. The signal from the undefective graphite lattice is the strongest for the S4 sample. For the three remaining samples, the share of the signal from the defective crystal lattice and amorphous carbon is increased to a similar degree (Table A2).

#### 2.1.2. Chemical Composition Analysis of GO Samples

The chemical composition of tested graphene oxides using the XPS technique indicated, in addition to expected carbon and oxygen, the presence of small amounts of contaminants. In the table below (see Table 1), each tested oxide’s chemical composition is combined with calculated atomic concentration.

The presence of carbon, oxygen, and small amounts of nitrogen, silicon, and sulfur was observed for each sample tested. Additionally, for one of the samples (S1), traces of aluminum were observed. Additional elements (N, Si, S) are present in small quantities, usually below one atomic percent. Only in the S3 sample, the atomic concentration of nitrogen was slightly higher and was at 1.36 at %. It was also observed that the ratio of atomic concentrations of carbon and oxygen is not the same for all tested samples. It is often at about 2.5, but for the S4 sample, this ratio is 17.89; a much higher amount of carbon and the significantly smaller amount of oxygen was detected.

Analysis of the sample’s constituents’ chemical state was based on the deconvolution of the C1s, O1s, N1s, S2p, and Si2p photoemission spectra. The results, including peak position and chemical state atomic concentration, were recalculated according to the total atomic concentration of a particular element and assignment of the particular chemical state combined in Table 2. The chemical state assignment was carried out on the basis of the NIST database [36].

Analysis of the high-resolution C1s spectra indicates carbon in several different chemical states, which is defined through the bond type. In examined graphene oxide, the C-C bond has been detected at a binding energy of about 284.8 eV; this type of bond is dominant in most of the samples, except for the S2 sample. The main contribution to the C1s spectrum gives the peak at 286.79 eV assigned to the C-O bond for that oxide. Such bonds have also been detected for the S1 and S3 samples. Additionally, the C=O bond at about 288.4 eV was detected in S1–S3 samples. In the S4 sample, the shape of the peak is different from that of other examined oxides. Bonds such as C=N at 284.03 eV, C-N at 286.11, C-OH at 285.44 eV, carbonyl bond >C=O at 287.06 eV, O-C=O bond at 289.03, and HO-C=O or π–π* satellite at 291.56 eV have been observed. The percentage contribution of each carbon with a specific bond, together with the atomic concentration computed from values given in Table 1, have been combined in Table A3.

Analysis of negative and positive secondary ion mass spectra confirmed the presence of detected with the use of XPS elements (C, O, Si, S, N, Al), additionally due to higher detection limits of the TOF-SIMS technique H, Na, Mg, F, Cl, K, and Ca were detected. Those elements are present in a small amount can be interpreted as surface contaminants. For the S2 and S4 oxides, there was relatively high Cl^-^, and for the S4 sample, a high Na^+^ signal was observed, suggesting that those samples were contaminated to a greater extent than the S1 sample (Figure 4(A1,A2)).

The mass spectra indicate several types of molecular species, namely C_x_^+^, C_x_^−^C_x_H_y_^+^, C_x_H_y_^−^, CO_x_^+^, CN^−^,SO^-^, NH_x_^+^, C_x_H_y_O_z_^+^, C_x_H_y_O_z_^−^, C_x_N_y_O_z_^+^,C_x_N_y_O^−^, C_x_H_y_N_z_^+^, and C_x_H_y_NO^+^. The intensities of secondary ions in S1–S3 secondary ion mass spectra are similar. Secondary ion mass spectra of the S4 sample show different peak patterns; even if the same secondary ions are present, the relative intensity ratios differ from other examined oxides (Figure 4(A3)).

The analysis of suspensions stability evaluations showed that the zeta potential of all GO samples was relatively high, amounting from −36.60 and −39.38 mV. It was an indication of considerable electrical stability of the suspensions (Figure 3(B1)).

### 2.2. Cytotoxicity Analysis

A ten-day experiment revealed that cell viability, the intensity of oxidative stress, the apoptosis level, and DNA damage in the cells of *A. domesticus* were dependent on nanoparticles’ concentration in the food and GO ingested type. All GO samples modulated cellular parameters with varying degrees of intensity. Moreover, in both concentrations, the GO samples showed different trends depending on the experiment’s time point across all parameters.

#### 2.2.1. Cell Viability and Oxidative Stress

Flow cytometry showed a statistically significant decrease in intestinal cell viability in *A. domesticus* in the groups treated with the highest concentration of nanoparticles in food (20 µg·g^−1^ of food). This relationship applied to all experimental groups but to a different extent. The S4 group showed the strongest response among all tested groups (20 µg·g^−1^ of food), and it was significantly different from the S2 group for the 10 days of the experiment (Figure 5(A2)). All GO groups correspond to the insects from the control; however, minor differences between the tested nanoparticle samples were noted (Figure 5(A1)). The Muse™ Oxidative Stress kit showed that samples S3 and S4 were the strongest in increasing the number of free radicals in the intestine at a lower concentration (2 µg·g^−1^ of food). In addition, sample S4 was characterized by a twice higher potential to generate free radicals than the other GO samples on the 10th day of the experiment (20 µg·g^−1^ of food) (Figure 5(B1,B2)).

#### 2.2.2. Apoptosis Level in the Cells

Muse™ Annexin V and Dead Cell assay showed that nanomaterials S2–S4 induced apoptosis strongly than the S1 at both tested concentrations (Figure 6(A1,A2)). Sample S4 differs significantly from the remaining 6 and 10 days of the experiment, where cells in early apoptosis were the most numerous (Figure 6(B1,B2)). The groups treated with the S1 sample showed cells with a significant late-phase apoptosis level, especially during 2 days of the experiment. The S2–S4 samples in food induced an increased degree of early and late apoptosis, but the groups did not differ significantly from the control group (Figure 6(C1,C2)).

#### 2.2.3. DNA Damage

The tested nanoparticles induce a genotoxic effect in insect hemocytes but with a different intensity depending on the time point and the DNA damage parameter measured (Figure 7). The percentage of tail DNA (TDNA) and tail length (TL) in nanometers were used as a parameter to describe DNA damage. The TDNA parameter was significantly higher in the first 6 days of the experiment in groups S1 and S3, and then, it returned to the control level on the last day of the experiment (2 µg·g^−1^ of food) (Figure 7(A1)). Organisms exposed to S2 and S4 materials in the higher concentration were characterized by a significantly higher TL (tail length) parameter from the 6th day of the experiment and remained at a similar level, also on the last day of the experiment (Figure 7(B2)).

## 3. Discussion

The literature on nanotechnology abounds with reports describing both the positive and negative effects of GO nanoparticles on various species of organisms [37,38,39,40,41]. In some respects, this study is a different view on the nanotoxicity aspect. The tested GO materials are by definition the same type of nanoparticles, and their effect on the model organism should be similar. However, considering the high reactivity of GO and the importance of physicochemical properties, we assumed that the finished GO products would affect the body with varying degrees, and the mechanism of their toxicity may be different. Physicochemical analyses revealed differences in properties. Many factors may be responsible for the difference in the same types of nanoparticles—for example, the synthesis methods, other methods of purifying the final product, or accidental contamination of the reaction mixture [42,43]. Raman spectroscopy showed that nanoproducts S1, S2, and S4 have a much less ordered crystal lattice structure and contain defective carbon layers characteristic of carbon black. The structure of the S4 material resembled polycrystalline graphite (Figure 2(A2,A3)). The visual examination of the samples prepared for dispersion in water (S1, S2, S4) led to the assumption that the suspensions will be characterized by a different size of flakes and a different potential for the formation of homogeneous solutions (Figure 8).

Presumed differences in the morphology of the samples were confirmed by scanning electron microscopy and atomic force microscopy. Samples S2 and S3 represented well-defined flake structures; sample S4 formed thick, blocky forms, while S1 tended to form three-dimensional aggregates of nanometric flakes (Figure 1(A2–A4)). S2 and S3 were small single-layer flakes, while sample S1 formed aggregates up to several dozen nanometers, and S4 even up to 1 µm (Figure 1(B1–B3)). Any change in the nanoparticle properties (composition and morphology) may reduce or increase their solubility in the dispersing phase (water, biological environment) and, going further, affect their reactivity and toxicity to organisms [44].

Considering the PCA analysis of biomarkers and physicochemical parameters, principal component 1 (PC 1) explained as much as 78.99% of the overall variability and visibly identified two clusters. The first cluster included data for early apoptosis, total apoptosis, oxidative stress, the tail length of the DNA comet, and most of the chemical and structural properties of the GO. The second cluster (markedly different) comprised data for living cells, the level of contamination of samples, and C=O, C-O groups in the GO surface. Principal component 2 (PC 2) explained 7.27% of the total variability and illustrated the third cluster gathering data for late apoptosis and tail DNA parameters. Visualization of the data on a 2D PCA plot allows for the separation of GO samples regarding their biological effect (Figure 9(A2)) and physicochemical properties (Figure 9(A3)). The most physicochemically different sample S4 (Figure 8(A3)) caused the most different biological effect (Figure 9(A2)). In addition, sample S1 differed from the others in terms of morphology/chemical properties (Figure 9(A3)). This GO sample also had a markedly different effect on biological markers (Figure 9(A2)).

### 3.1. Effect of GO Samples on Cell Viability and Parameters of Apoptosis

All GO products, with increasing exposure time and concentration, significantly decreased the viability of *A. domesticus* intestinal cells (Figure 5(A1,A2)). The effect of increased cell death was most visible in the case of contact of the animals with S4 and S1 materials at the concentration of 20 µg^−1^ (Figure 5(A2)). The results indicate that induced apoptosis was of great importance in the process of cell death. S4 nanoparticles significantly enhanced apoptosis on the 6th day of the experiment at both tested concentrations (Figure 6(A1,A2)). The S4 and S3 materials contributed most to the increase in the number of cells in the early apoptosis phase. In contrast, the S1 material caused the increase in late apoptosis already on the second day of the experiment in both concentrations (2 µ·g^−1^ and 20 µ·g^−1^). This tendency was maintained at the highest concentration until the 10th day of the study (Figure 6(C1,C2)). Early apoptosis is defined as an increase in phosphatidylserine lipid expression, detected by annexin V on an intact cell membrane. Late apoptosis is characterized by a loss of integrity through the cell membrane [45]. We can assume that the S1 and S4 materials from all tested GO samples are the most effective in the breakdown of cell membranes. The mechanism of these particles’ toxicity is related to their large size and tendency to aggregate (Figure 1(A1,A4)) (Figure 9(A1)). This hypothesis does not coincide with the results of studies that associate lower toxicity with an increase in nanoparticles’ size [46], but the obtained results are justified by the mechanism of the internalization of nanoparticles with the cell by endocytosis. Small sheets of graphene oxide enter cells through clathrin-dependent endocytosis, while larger ones enter via phagocytosis [47,48,49]. Nanoparticle aggregates are sufficient to induce appropriate reactions on the cell membrane’s surface and activate the transport process. It depends on the type of charge on their surface and is related to, for example, the density of ligands (functional groups) on which the thermodynamic driving force of endocytosis depends [50]. In the case of small GO flakes, there may be times when the number of ligands is insufficient to accomplish the “invasion” process. A similar remark also applies to large structures. The endocytosis process initiation can be triggered either by the clustering of many small nanoparticles on the surface of the cell membrane or by the high density of functional groups on the surface of large structures that act as crosslinking agents capable of aggregating nearby receptors [51]. Moreover, massive reactions of particles with the cell membrane can contribute to the production of reactive oxygen species and generate oxidative stress responsible for the breakdown of membrane lipids, changes in metabolic pathways, mitochondrial and nuclear DNA dysfunction, and finally, cause apoptosis or necrosis of the cell [52].

### 3.2. Effect of GO Samples on the Level of Oxidative Stress

Interactions between particles and cells play a key role in generating oxidative stress [53]. The free radicals formed can either be directly bound to the GO surface or be included as free units in the dispersing medium (cell environment), especially when the nanoparticle has good solubility and many oxygen functional groups [44,54]. Oxidative stress arises through an impaired electron transport chain, structural damage to cell organelles (including the cell membrane), or depolarization of the mitochondrial membrane [55,56,57]. It is considered that the potential of a nanoparticle to generate oxidative stress is correlated with its size, which determines the number of reactive groups on the surface and its high reactivity ratio. Reduced particle size usually results in structural defects and altered electrical properties. However, it does not have to. Nanoparticles of identical shape and size can lead to diverse cytotoxic responses due to their surface properties, and metals such as iron (Fe), copper (Cu), chromium (Cr), vanadium (V), and silica (Si) bound to the GO surface can accelerate ROS formation through Haber–Weiss and Fenton reactions [54,58]. The obtained results showed that all tested GO materials (S1–S4) in a concentration-dependent manner contributed to the overproduction of free radicals, leading to an antioxidant imbalance in gut cells (Figure 5(B1,B2)). The statistical analysis showed that the materials S4 and S1 initiated the strongest oxidative stress at the concentration of 20 µg^−1^ and significantly differed from the remaining nanoparticles (Figure 5(B2)) (Figure 9(A2,A3)). As mentioned, these materials had a high aggregation potential, and flakes were of considerable size after sonication. S1 and S4 had lower surface energy and less ionizable surface groups, which enhance hydrophilicity [59]. Therefore, the mechanical interaction with biological structures may be more likely as the mechanism of their toxicity than the release of free radicals in the cell’s environment. However, a change in surface charge density to a more negative effect in the cell’s environment and dissociation of hydroxyl and carboxyl groups due to ionization cannot be excluded. Moreover, the S4 material was a form of polycrystalline graphite, which influenced the functionalization of its surface, a lower degree of oxidation, confirmed by XPS analysis (Table 1 and Table 2), and the presence of impurities with the metal-containing functional groups (Figure 9(A1–A3)). As indicated by PCA analysis, the ten-day lasting exposure to contaminants in the samples could lead to cell viability stimulation. The reason for the observed relationship may be the activation of compensatory mechanisms contributing to the hormesis effect (Figure 9(A1)). Another likely scenario is the higher susceptibility of nanoparticles to reducing digestive enzymes’ environment and organic components in the food. The S1–S3 materials showed a higher crystal lattice defect compared to the S4 material (Table A2), but their planes could be rearranged, and as a result of the reduction of oxo groups, if the C-C bonds were not lost, their structure was restored. Structure defects have become temporary [13]. The incomplete reduction process in the biological environment may result in a significant number of surface defects. In the sample, S4 could intensify the tendency to disturb cell membranes’ integrity, including the mitochondria, leading to the generation of free radicals causing damage to the genetic material [53,57].

### 3.3. Effect of GO Samples on the Genetic Material of the Cell

Genotoxicity can be initiated either in the primary (direct interaction of the nanoparticle with genetic material) or in a secondary way, through the destruction of DNA due to increased oxidative stress, structural DNA mutations, cell cycle genes, and dysfunction of intracellular information transmission pathways [60]. The smallest particles (up to 10 nm) can pass through the nuclear membrane by diffusion or through numerous nuclear pore complexes and interact directly with DNA. Larger ones get inside the nucleus during cell division (mitosis) when the nuclear membrane disappears [61]. DNA damage in the tested GO may have resulted from the combination of a chain of events of primary and secondary mechanisms. The TDNA parameter showed that at a lower concentration of the sample, S1 and S3 contributed to the increase in DNA damage on the 3rd and 6th day of the experiment, which returned to the control level at the end of the experiment (Figure 7(A3)). At a higher concentration, the genotoxicity of samples S2 and S4 increases on day 6 and remains at a significantly higher level until the end of day 10. Taking into account the morphology and structure of the S4 sample, we can assume that the DNA damage appeared as a result of increased oxidative stress and, as a consequence, significant damage to the DNA of the cells was directed to the apoptotic pathway (Figure 6(A2,B2)), (Figure 9(A1)). S2 nanoparticles (GO single-layer flakes, 1 nm) could interact with the cell both with free radicals’ participation and damage DNA in direct contact after penetrating the nuclear membrane. The concentration of nanoparticles was significant. Genome integrity is strategic for the cell, and many repair mechanisms prevent genetic information loss. The lower concentration of nanoparticles caused minor defects that could be repaired quickly without harming the body. Further analyses are needed to determine whether the S1 and S4 materials during long-term exposure will trigger repair enzymes, lead to mutations causing cell death, or transform into cancer cells.

## 4. Materials and Methods

### 4.1. Experimental Model

The experiment consisted of two stages. First, four GO nanoparticle suspensions were prepared and characterized using a multi-method approach, including microscopic and spectroscopic techniques. Then, each type of GO was mixed with the ground food and given to animals during the cytotoxicity assessment. A detailed food preparation method and the food composition can be found elsewhere [26,27].

The insects used for the experiment were obtained from a laboratory stock population maintained at the University of Silesia in Katowice (Poland). The adults *A. domesticus* (about 28 days of imago stage) were divided into experimental groups, including a control group (C) and two contaminated groups with GO: 2 μg·g^−1^ of food and 20 μg·g^−1^ of food for each of the nanoparticle suspension. A total of 12 experimental groups were bred in separate plastic boxes (24 cm × 16 cm × 16.5 cm, 30 individuals per each) with access to shelter, food, and fresh water. The insects were given food with GO nanoparticles for 10 days. Randomly, five animals were taken from each group at selected time points (2, 6, and 10 days of exposure) to analyze stress parameters in the gut or hemolymph of insects, depending on the assay (Scheme 1).

### 4.2. GO Suspensions Preparation

Graphene oxide was purchased from various companies: Sigma Aldrich, USA (Graphene oxide powder; 15–20 sheets), ACS Material, USA (Single Layer Graphene Oxide; H method), MSE PRO, USA (Monolayer Graphene oxide powder), and Nanografi, USA (Graphene oxide dispersion in water; 10 mg·mL^−1^). For this article’s purposes, nanoparticle samples were coded as samples 1–4 (S1–S4).

GO stock solutions (10 mg·mL^−1^) added to the food were prepared by ultrasonic homogenization (cycle: 1 amplitude, 100%; model UP-100H, DONSERV) of mixed GO flakes in ultrapure water. The sonication of each GO was performed under the same conditions (10 h, on ice) to obtain a homogeneous mixture of nanoparticles. Graphene oxide obtained from Nanografi was used in the experiment in the form of a manufacturer’s solution. Final products were stored tightly closed in the dark at room temperature.

### 4.3. Physicochemical Characterization of GO

Morphological and structural analyses: The morphology and structure of samples were characterized by scanning electron microscopy (SEM), atomic force microscopy (AFM), transmission electron microscopy (TEM), and Raman spectroscopy.

For the analysis, highly diluted GO dispersions with a concentration to 8 µg·mL^−1^ were deposited on the silicon wafer and fresh cleaved mica for SEM and AFM measurements, respectively [62]. Samples were left for drying at room temperature [63]. SEM analyses were realized at high vacuum conditions, with beam accelerating voltage of 2 kV, to ensure good contrast even during single-layered GO imaging. The hard tapping mode cantilever All-In-One Al (Budget Sensors) probe D, with a typical force constant of 40 N/m and resonant frequency 350 kHz, was used for AFM measurements (Agilent 5500, Agilent technologies, Santa Clara, CA, USA). The standard scan frequency was 0.2 Hz [35,62,64].

Samples for TEM observation were prepared by dispersing a small amount of the material in ethanol and putting a droplet of the suspension on a microscope copper grid covered with carbon film and allowed to evaporate the alcohol. Then, samples were dried and purified in a plasma cleaner. TEM investigations were done on a probe Cs-corrected S/TEM Titan 80–300 FEI microscope, equipped with energy dispersion X-ray spectrometer (EDS).

The Raman spectra (Renishaw) were collected at room temperature in the spectral range from 800 to 3500 cm^−1^ using a 20 mW argon laser (wavelength λ = 514.5 nm). The spectra were obtained with a 5% value of the laser beam power and an exposure time of 60 s. For each sample, three spectra were determined at different locations on the sample. For all GO samples, the first-order Raman spectra display two main peaks (Figure 3(A2)), which are characteristic features of graphitic carbons [65]. The peak at around 1580 cm^−1^ (called graphitic or G band) is assigned to the C=C stretching vibration mode and observed in aromatic rings of graphitic structures. The D peak at around 1350 cm^−1^ (so-called disorder mode) is activated by defects and not observed for single-crystal perfect graphite. The origin of the D mode is attributed to the double-resonant Raman scattering [66]. The ratio of both bands intensities ID/IG can be used to calculate the averaged in-plane (measured in the direction of the carbon layers) size of graphitic domains (La) from the formula of Knight and White [67], based on Tuinstra and Koenig [65].
$$L_{a}=4.4{\left(\frac{I_{D}}{I_{G}}\right)}^{-1}$$
where *I_D_* and *I_G_* denote the intensities (area under the curve) of the *D* and *G* bands, respectively. For detailed analysis of Raman spectra, the method proposed by Sadezky (deconvultion of experimental spectra into five bands: four Lorentz and one Gauss) was used [68]. Table A1 presents the position of the Raman bands along with the interpretation of their source. The calculations were made with the use of the FITYK program [69].

Surface analysis: The elemental composition of the GO surface was determined using X-ray photoelectron spectroscopy (XPS) PHI 5700 (Physical Electronics Inc., Minnesota, USA) and time of flight–secondary ion mass spectrometry (TOF-SIMS) (ION-TOF GmbH, Munster, Germany) [70,71]. The electrokinetic potential (ζ potential) of the GO suspension (in DI water) was measured at 25 °C using a ZetaSizer Nano ZS system (Malvern Instruments, Malvern, UK) [72].

The XPS measurements were executed using a monochromatic Al Kα radiation of 1486 eV energy to excite photoelectrons from the sample surface [73]. The photoemission spectra were collected in a wide binding energy range (−2–1400 eV) and high-resolution narrow ranges. The narrowed ranges represent the characteristic photoemission lines of individual elements detected on the surface of examined graphene oxides. The analysis was carried out using PHI MultiPak (v.9.6.0.15) software. The XPS peaks were fitted to Voigt functions after performing a Shirley background subtraction. XPS has allowed obtaining information about the samples’ chemical composition with the accuracy of detection of 0.1 atomic percent from the surface with a thickness of about 3 nm.

Further analysis was realized with the TOF-SIMS instrument equipped with a bismuth liquid metal ion gun of the energy of 30 keV and a current of about 1 pA. Positive and negative secondary ions spectra were collected by rastering the ion beam across 500 × 500 µm areas. The analysis was carried out using SurfaceLab 6 software. The positive spectra were calibrated using CH_3_^+^, C_2_H_3_^+^, C_3_H_3_^+^, C_3_H_4_^+^, and C_3_H_5_^+^ ions, whereas negative one using C^−^,CH^−^, C_2_^−^, C_2_H^−^, C_3_^−^, and C_3_H^−^. TOF-SIMS allowed more accurate chemical analysis than XPS (it is associated with a higher detection limit and a smaller sampling depth of about 0.4 nm of the TOF-SIMS technique) [74].

### 4.4. Cytotoxicity Evaluation

The cytotoxicity of nanoparticles in the insect’s gut was assessed with Muse^®^ Cell Analyzer (Muse^®^ Cell Analyzer; Millipore, Billerica, MA, USA), which uses fluorescence detection and microcapillary cytometry to deliver quantitative single-cell analysis. A cell suspension was prepared by shaking the tissue with a 0.1 M phosphate buffer (pH 7.4) in a homogenizer (Minilys, Bertin Technologies, Montbonnot, France). Determination of early/late cell apoptosis, the proportion between alive and dead cells, and the level of reactive oxygen species (ROS) were performed based on the manufacturer’s diagnostic kits: Muse^®^ Annexin V and Dead Cell Kit and Muse^®^ Oxidative Stress Kit assay, respectively.

Genotoxic effect of GO was measured in hemolymph suspension (in a 1:1 ratio hemolymph and anticoagulant buffer solution) by Single Cell Gel Electrophoresis assay (SCGE). Briefly, 180 slides (total in the study) were incubated with lysis solution (90 min) and electrophoresed, using an alkaline buffer (4 °C, 20 min at 0.3 A, 1000 mL). Finally, the slides were submerged in a neutralization solution and dehydrated in methanol of 99.8%. Slides were stained with DAPI dye and visualized under a fluorescent microscope with Komet 5.5 image analysis system (Kinetic Imaging, Liverpool, UK). More details about comet assay are presented in our previous article [75,76].

All the tests were performed at least five times for each time point in the experimental group.

### 4.5. Statistical Analysis

Data normality for all biological parameters was tested using the Kolmogorov–Smirnov and Shapiro–Wilk tests. Levene and Brown–Forsythe tests confirmed the homogeneity of variances. Therefore, significant differences were determined by the honest significant difference test (HSD Tukey test, ANOVA; *p* < 0.05). Principal Component Analysis (PCA) for evaluating the relations among all biological (measured at 10th day of exposure to GO at the concentration of 20 μg·g^−1^) and physicochemical parameters was performed. Statistical analysis was prepared using Statistica 13.1.

## 5. Conclusions

The study proved that the tested GO suspensions have different potential to induce cytotoxicity (Figure 9(A2)), and the mechanism of interaction with the cell is difficult to standardize and define. After 6 days of exposure, crucial physiological responses of various nature appeared, including repair mechanisms, which strongly correlated with the properties of GO and the concentration of nanoparticles in the food.

Even subtle differences in the chemical composition and morphology of GO can lead to unforeseen reactions in the organism’s environment. The biocompatibility of GO is a small and susceptible area that can be presented as a resultant of many factors derived from the GO and the organism/cells (Scheme 2). Since GO is a future-oriented platform, its in-depth assessment and characterization of the potential to react with biological material are highly recommended. To explain individual physiological responses and set toxicity standards, we plan more precise analyses.

## Data Availability

Data is contained within the article or supplementary material.

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
