# Peer review of "The Structure–Properties–Cytotoxicity Interplay: A Crucial Pathway to Determining Graphene Oxide Biocompatibility"

_ijms, 2021, doi:10.3390/ijms22105401_

Round 1

Reviewer 1 Report

The manuscript entitled "The structure-properties-cytotoxicity interplay - a crucial pathway to determining graphene oxide biocompatibility" has great potential to the readers in which authors reported graphene oxide biocompatibility. This study can be published in IJMS after some revisions.

  1. The manuscript lacks references, thus authors are advised to provide an authentic reference for each claiming statement within the introduction and throughout the manuscript (where applicable).
    Some recommendations are;
    Introduction section
    https://doi.org/10.1016/j.molliq.2017.08.129
    XPS analysis
    https://doi.org/10.3390/polym13030478
    Similarly please find appropriate references and put with respect to the reason of research results in each characterization section.
  2. Please fix the lines that are merged in table 1. 
  3.  Keep graphs vertically in figure 8 to maintain the manuscript format as per mdpi standards.
  4. On line 521, the Authors should be specific with the values by avoiding (few) in the sentence  "about a few µg per ml"
  5. Please provide a reference or reason why tapping mode was chosen for AFM analyses.
  6. There should be a gap between the reference bracket and the ending word within the text e.g. On line 33-34, in sentence materials are more environment friendly than organic material [1]. not material[1]. On Line 53 in the literature [11-13].
  7. Figure 1. parts from B1-B4 are not discussed in respective section 2.1.1.  Also, please elaborate the reasons of the outcomes!
  8. The quality of Figure 3 should be improved, fonts are almost invisible.
  9. The authors should explain why this study is related to in vivo?
    Otherwise, authors are suggested to delete the word "in vivo" from the Keywords.
  10. The overall article language is very good, the authors are advised to correct a few remaining typos within the manuscript such as missing articles and "." full stop sign is missing at many places within the manuscript. Also, cm-1 should be replaced by cm-1, CO2  to CO2, additionally authors are encouraged to find typos and correct them.

Author Response

Dear Reviewer,

Thank you very much for your kind reviews and valuable remarks and suggestions. They were very useful for the preparation of the corrected version of our manuscript. All of them have been taken into account and we believe that considerable and satisfying changes were introduced into the corrected version of the manuscript. Below are our replies to your points.

Point 1: The manuscript lacks references, thus authors are advised to provide an authentic reference for each claiming statement within the introduction and throughout the manuscript (where applicable). Some recommendations are;

Introduction section

https://doi.org/10.1016/j.molliq.2017.08.129

XPS analysis

https://doi.org/10.3390/polym13030478

Similarly please find appropriate references and put with respect to the reason of research results in each characterization section.

Response 1: Thank you for the suggestion. In the corrected version of the manuscript, the missing references were added.

Point 2: Please fix the lines that are merged in table 1.

Response 2: The correction was made. We followed your suggestion.

Point 3: Keep graphs vertically in figure 8 to maintain the manuscript format as per mdpi standards.

Response 3: We did as you suggested.

Point 4: On line 521, the Authors should be specific with the values by avoiding (few) in the sentence  "about a few µg per ml"

Response 4: We added missing information.

Point 5: Please provide a reference or reason why tapping mode was chosen for AFM analyses.

Response 5: Graphene Oxide (GO) analysis was realized using Atomic Force Microscopy (AFM) technique. The standard procedure consists of dispersions' deposition on a selected substrate, sample drying and AFM imaging. Firstly the proper substrate should be chosen. The typical ones are mica, silicon oxide, less often HOPG. The best choice can be here the mica due to its hydrophilicity, it ensures also a better quality of GO imaging by the lower roughness, which follows into GO flakes morphology [1]. The drying was realized by room temperature evaporation to prevent any temperature changes, mostly the GO reduction, which can start even in a relatively low temperature of 70°C [2]. Finally, the AFM imaging can be realized in two different modes: the contact mode and the tapping mode (also known as AC mode). In the case of GO AFM imaging, both of them are appropriate. However, the tapping mode ensures much softer interaction between tip and sample. Thus, tapping mode a more common choice [1, 3, 4]. The last parameter influencing AFM measurement is the GO dispersion concentration. The resulting surface coverage should be below one monolayer, which ensures good flakes recognition. The appropriate concentration should be chosen experimentally. In our work, we diluted the initial GO water dispersions down to 8 µg/ml by the ultra-pure Milli-Q water, to prevent any external contamination. The same diluted dispersions were used for SEM analysis but deposited on the silicon substrate.

The references below have been provided into the text:

[1] Wilson, N. R., Pandey, P. A., Beanland, R., Rourke, J. P., Lupo, U., Rowlands, G., & Römer, R. A. (2010). On the structure and topography of free-standing chemically modified graphene. New Journal of Physics, 12(12), 125010. https://doi.org/10.1088/1367-2630/12/12/125010

[2] Jung, I., Field, D. A., Clark, N. J., Zhu, Y., Yang, D., Piner, R. D., … Ruoff, R. S. (2009). Reduction Kinetics of Graphene Oxide Determined by Electrical Transport Measurements and Temperature Programmed Desorption. The Journal of Physical Chemistry C, 113(43), 18480–18486. https://doi.org/10.1021/jp904396j

[3] den Boer, D., Weis, J. G., Zuniga, C. A., Sydlik, S. A., & Swager, T. M. (2014). Apparent Roughness as Indicator of (Local) Deoxygenation of Graphene Oxide. Chemistry of Materials, 26(16), 4849–4855. https://doi.org/10.1021/cm502147f

[4] Arif, T., Colas, G., & Filleter, T. (2018). Effect of Humidity and Water Intercalation on the Tribological Behavior of Graphene and Graphene Oxide. ACS Applied Materials & Interfaces, 10(26), 22537–22544. https://doi.org/10.1021/acsami.8b03776

Point 6: There should be a gap between the reference bracket and the ending word within the text e.g. On line 33-34, in sentence materials are more environment friendly than organic material [1]. not material[1]. On Line 53 in the literature [11-13].

Response 6: Thank you. The correction was made.

Point 7: Figure 1. parts from B1-B4 are not discussed in respective section 2.1.1.  Also, please elaborate the reasons of the outcomes!

Response 7: Thank you. Of course, you are right! Figure 1. B2, B3 should be instead of Figure 2. Missing references to the remaining figures have also been added (Figure 1. B1, B4). This mistake was a result of the renumbering of figures.

Point 8: The quality of Figure 3 should be improved, fonts are almost invisible.

Response 8: We followed your suggestion. Figure 3 was divided into Figure 3 and Figure 4. We hope that now the quality is better and letters are visible.

Point 9: The authors should explain why this study is related to in vivo? Otherwise, authors are suggested to delete the word "in vivo" from the Keywords.

Point 9: In this study, A. domesticus was exposed to GO for 10 days with different concentrations.  We did not work in vitro (with cell culture) but in vivo (with animals). Tissue homogenates were freshly prepared from animals fed with GO-food. The nanoparticles were detailed characterized before being added to the food and consumption. Therefore, we hope that these studies will be helpful in future assessments of GO's behavior in the biological environment (organism). The explanation of experimental design is included in the Materials and Methods chapter.

Point 10: The overall article language is very good, the authors are advised to correct a few remaining typos within the manuscript such as missing articles and "." full stop sign is missing at many places within the manuscript. Also, cm-1 should be replaced by cm-1, CO2  to CO2, additionally authors are encouraged to find typos and correct them.

Point 10: Thank you for your comments. The correction was made.

Reviewer 2 Report

The manuscript entitled “The structure-properties-cytotoxicity interplay - a crucial pathway to determining graphene oxide biocompatibility” aims to determine and compare the in vivo toxicity potential of GO samples from various manufacturers. The authors did a great job addressing the very important theme in a detailed manner, providing the necessary information concerning GO. The manuscript was overall written very well. I would recommend it for publication after checking minor spelling errors and just one comment: in Introduction, line 36, there is the following statement “Therefore, it was expected that graphene, a single graphite layer and the common building element of other carbon structures, would also be safe and valuable for technological purposes, mainly due to its excellent mechanical, electronic, and optical properties [2,3].” Regarding this, I can't entirely agree; due to the change in the dimensions of the material, it would be expected that there is a change in the potential toxicity as well. I recommend the authors take this into account.

Author Response

Point 1: The manuscript entitled “The structure-properties-cytotoxicity interplay - a crucial pathway to determining graphene oxide biocompatibility” aims to determine and compare the in vivo toxicity potential of GO samples from various manufacturers. The authors did a great job addressing the very important theme in a detailed manner, providing the necessary information concerning GO. The manuscript was overall written very well. I would recommend it for publication after checking minor spelling errors and just one comment: in Introduction, line 36, there is the following statement “Therefore, it was expected that graphene, a single graphite layer and the common building element of other carbon structures, would also be safe and valuable for technological purposes, mainly due to its excellent mechanical, electronic, and optical properties [2,3].” Regarding this, I can't entirely agree; due to the change in the dimensions of the material, it would be expected that there is a change in the potential toxicity as well. I recommend the authors take this into account.

Response 1: Dear Reviewer, thank you very much for your positive words. We are happy to hear you have found our research interesting. We made sure to correct the manuscript according to your suggestion. We hope you will be satisfied with our changes to the manuscript.
